# Generation of Alzheimer’s Disease Model Derived from Induced Pluripotent Stem Cells with *APP* Gene Mutation

**DOI:** 10.3390/biomedicines12061193

**Published:** 2024-05-27

**Authors:** Yena Kim, Binna Yun, Byoung Seok Ye, Bo-Young Kim

**Affiliations:** 1Division of Intractable Disease Research, Department of Chronic Disease Convergence Research, Korea National Institute of Health, Cheongju 28160, Republic of Korea; kyena0430@gmail.com (Y.K.); dbsdudwn0921@korea.kr (B.Y.); 2Korea National Stem Cell Bank, Korea National Institute of Health, Cheongju 28160, Republic of Korea; 3Department of Neurology, Yonsei University College of Medicine, Seoul 03722, Republic of Korea; romel79@yuhs.ac

**Keywords:** Alzheimer’s disease, amyloid precursor protein, cerebral organoid, disease modeling, drug screening, induced pluripotent stem cells, organoids, tau pathology

## Abstract

Alzheimer’s disease (AD), the most common cause of dementia, is characterized by disruptions in memory, cognition, and personality, significantly impacting morbidity and mortality rates among older adults. However, the exact pathophysiological mechanism of AD remains unknown, and effective treatment options for AD are still lacking. Human induced pluripotent stem cells (iPSC) are emerging as promising platforms for disease research, offering the ability to model the genetic mutations associated with various conditions. Patient-derived iPSCs are useful for modeling neurodegenerative and neurodevelopmental disorders. In this study, we generated AD iPSCs from peripheral blood mononuclear cells obtained from a 65-year-old patient with AD carrying the E682K mutation in the gene encoding the amyloid precursor protein. Cerebral organoids derived from AD iPSCs recapitulated the AD phenotype, exhibiting significantly increased levels of tau protein. Our analysis revealed that an iPSC disease model of AD is a valuable assessment tool for pathophysiological research and drug screening.

## 1. Introduction 

The brain comprises a complex architecture that regulates the functions of several organs. It presides over essential cognitive processes, such as thought, decision making, memory, emotional responses, movement, speech, respiratory control, temperature control, and the regulation of organ function. A wide range of diseases and disorders affect the brain, including brain tumors, neurodegenerative diseases, neurodevelopmental disorders, encephalitis, stroke, and traumatic brain injury. Alzheimer’s disease (AD) is an irreversible chronic disease that affects the brain and slowly destroys memory, cognition, and personality. The hallmark of AD is the accumulation of amyloid beta (Aβ) and tau proteins [1]. Familial AD, an autosomal dominant inherited form of AD, is caused by mutations in the *APP* gene, encoding the amyloid precursor protein (APP), presenilin 1, and presenilin 2. Approximately 70% of the risk of developing AD can be attributed to genetic factors [2]. In particular, *APP* gene mutations influence the Aβ levels and disease risk. 

Globally, more than 55 million people have dementia, which is the leading cause of physical and cognitive disabilities [3]. Its incidence, prevalence, and mortality rates have increased dramatically [4]. Currently, no cure, medications, treatments, or other therapies are available for AD. Challenges in distinguishing between human brain physiology and disease model systems have hindered both therapeutic development and the understanding of the disease’s pathology [5]. Furthermore, biochemical and physiological differences exist between human and animal models. 

One promising strategy for overcoming the scarcity of biomaterials in diseases is to generate induced pluripotent stem cells (iPSCs) [6]. Human iPSCs (hiPSCs) are obtained by reprogramming somatic cells (e.g., cord blood mononuclear cells, peripheral blood mononuclear cells (PBMCs), fibroblasts, keratinocytes, and hair follicle cells) using lentiviruses, Sendai viruses, or episomal vectors. HiPSCs are pluripotent and can differentiate into three germ layers in vitro. Patient-derived iPSCs are valuable tools for modeling brain diseases and can differentiate into disease-related cells [7,8]. Furthermore, iPSC-based systems and genome-editing tools will be critical in understanding the roles of numerous newly discovered genes and mutations that influence disease risk. 

Many protocols and methods have been developed to differentiate iPSC-derived neural models [9]. Two-dimensional (2D) methods are relatively simple and require a short differentiation period. Three-dimensional (3D) methods, such as spheroids and organoids, offer a more functionally complex system that mimics the developmental process. Cerebral organoids (COs) derived from iPSCs are self-assembling 3D cellular aggregates that mimic the human brain. CO models have emerged as a new approach for investigating human brain diseases in vitro, overcoming the conventional limitations of in vitro human-cell-based systems and animal models [10]. CO models facilitate studying pathological mechanisms and screening for potential drugs. Many hiPSC-derived brain organoid models have been developed to study brain diseases [11]. However, 3D methods require prolonged culture periods, ranging from 80 days to 1 year. Such extended culture times are constrained by both time and cost considerations. In addition, prolonged culture durations often lead to necrosis within the organoids [12]. 

This study generated an iPSC line from PBMCs obtained from a patient with AD carrying the E682K mutation in the *APP* gene. Furthermore, we describe a shortened differentiation protocol for COs, facilitating the efficient disease modeling of AD. 

## 2. Materials and Methods

### 2.1. Reprogramming of PBMCs Derived from Patient with AD and Maintenance of iPSCs

The iPSC control line (HC) was established in our group by reprogramming healthy human peripheral blood mononuclear cells using the CytoTune-iPS 2.0 Sendai reprogramming kit (Thermo Fisher Scientific, Waltham, MA, USA). AD was an iPSC line generated from the PBMCs of a patient harboring a mutation in the *APP* gene: NM_000484.3: c.2044TG>A (p.E682K). Clinical information for both the patient and a healthy control is provided in Table 1. The generated iPSCs were maintained under feeder-free conditions on iMatrix-511–coated (Nippi, Tokyo, Japan) dishes containing StemFit Basic 03 medium (Ajinomoto, Tokyo, Japan) with 100 ng/mL of bFGF (Sigma-Aldrich, St. Louis, MO, USA). 

### 2.2. Alkaline Phosphatase Staining and Immunofluorescence Staining of AD iPSCs

Alkaline phosphatase staining was performed according to the manufacturer’s instructions. Briefly, the iPSCs were fixed with 4% paraformaldehyde (Fujifilm Wako, Osaka, Japan) for 3 min and incubated with the staining solution for 15 min in the dark. The stained iPSC colonies were examined using bright-field microscopy.

For immunofluorescence staining, the iPSCs were fixed with 4% paraformaldehyde (Wako) for 20 min and permeabilized with 0.2% Triton X-100 (Sigma-Aldrich. St. Louis, MO, USA) for 10 min. The cells were blocked for 30 min in 1% bovine serum albumin (BSA; Sigma-Aldrich, St. Louis, MO, USA) and incubated with primary antibodies. The cells were treated with the secondary antibodies for 1 h; counterstaining was performed with 4′,6-diamidino-2-phenylindole (DAPI). The images were analyzed using fluorescence microscopy (Olympus JP/IX83, Tokyo, Japan). 

### 2.3. Real-Time Polymerase Chain Reaction (RT-PCR) Analysis of AD iPSC

The total RNA was isolated from the AD iPSCs using the Maxwell RSC simplyRNA Cell Kit (Promega, Madison, WI, USA) and transcribed to cDNA using EcoDry Premix (Clontech, Kusatsu, Shiga, Japan). RT-PCR was performed using the TaqMan gene expression Master Mix (Applied Biosystems, Waltham, MA, USA). The TaqMan probe IDs are described in Table 2. The gene expression levels were normalized to the *GAPDH* levels for standardization. 

### 2.4. Mycoplasma Test of AD iPSC

The cell supernatants were analyzed using the e-Myco™ VALID Mycoplasma PCR Detection Kit (25239; Boca Scientific, Dedham, MA, USA) according to the manufacturer’s instructions. The amplified products were analyzed via gel electrophoresis. 

### 2.5. Karyotyping of AD iPSC

The karyotypes were determined by standard cytogenetic procedures using the GTG-band method. The cells were treated with colcemid for 45 min, incubated in a hypotonic solution, and fixed with a methanol–acetic acid solution (3:1). After Giemsa trypsin banding, the karyotypes were analyzed according to the International System for Human Cytogenetic Nomenclature using the standard G-banding method. 

### 2.6. In Vitro Three Germ Layer Differentiation of AD iPSC

For three germ layer differentiation in vitro, the hiPSCs were induced to form embryoid bodies (EBs) using Dulbecco’s Modified Eagle Medium F-12 (Thermo Fisher Scientific, Waltham, MA, USA) with 20% knockout serum replacement (Thermo Fisher Scientific, Waltham, MA, USA), 1% nonessential amino acids (Thermo Fisher Scientific, Waltham, MA, USA), and 0.1 mM β-mercaptoethanol (Gibco, Waltham, MA, USA) for 14 days. The differentiated EBs were analyzed via RT-qPCR using the TaqMan gene Expression Master Mix (Applied Biosystems, Waltham, MA, USA). The TaqMan probes are listed in Table 2. The gene expression levels were normalized to the *GAPDH* levels for standardization. 

### 2.7. Sequencing Analysis 

Sanger sequencing was used to detect mutations in the *APP* gene. Genomic DNA was extracted from the generated iPSCs using the Maxwell RSC Blood DNA kit (AS1400; Promega, Madison, WI, USA). Primers were designed to amplify variants using Primer3, and the primer sequences are described in Table 2. Amplicons were purified and sequenced in both directions using the BigDye Terminator Cycle Sequencing kit (4337456; Applied Biosystems, Waltham, MA, USA) and an ABI 3730XL Genetic Analyzer (Applied Biosystems, Waltham, MA, USA). 

### 2.8. Differentiation of hiPSC into Cerebral Organoids 

CO differentiation was performed using the STEMdiff Cerebral Organoid Kit (Stem Cell Technologies, Vancouver, BC, Canada) with some modifications. For EB formation, an AggreWell plate (34850; Stem Cell Technologies, Vancouver, BC, Canada) was pretreated with an anti-adherence rinsing solution (07010; Stem Cell Technologies, Vancouver, BC, Canada) and centrifuged at 1300× *g* for 5 min. Each well was rinsed with warm EB formation media, and the warm media was added to each well. The hiPSCs were detached using Accutase, resuspended in EB formation medium (9 × 10^3^ cells/EB) with 5 mM Y-27632, and transferred to an AggreWell plate. The AggreWell plates were incubated at 37 °C with 5% CO_2_. On days 2 and 4, 1 mL of medium was removed and 1 mL of EB formation medium was added to each well. On day 5, the EBs were transferred to a six-well ultra-low attachment plate with induction medium and incubated at 37 °C for 48 h. On day 7, the EBs were transferred to an embedding surface (Parafilm) using a wide-bore 200 µL pipette tip, and the excess medium was removed from the EB. We added 15 µL of Matrigel (Stem Cell Technologies, Vancouver, BC, Canada) onto each EB, repositioned the EB to the center of the Matrigel, and incubated the culture at 37 °C for 30 min. The embedded EBs were transferred to ultra-low attachment plates containing induction medium and incubated at 37 °C for 3 days. On days 10–40, the organoid medium was replaced with maturation medium, and the cultures were incubated on an orbital shaker at 37 °C. 

### 2.9. Immunofluorescence

For cryosection, COs were fixed with 4% paraformaldehyde overnight and incubated with 0.1% Tween 20. The fixed COs were treated with 15% and 30% sucrose for cryoprotection. The COs were embedded in optimal cutting temperature compound and sectioned into 10 µm slices using a microtome. The slides were dried for 30 min at room temperature and washed with 1× phosphate-buffered saline (PBS). The slides were blocked with 5% BSA in PBS with 0.5% Triton X-100 and incubated overnight with primary antibodies against TUJ1 (18207; Abcam, Cambridge, UK), MAP2 (13-1500; Invitrogen, Waltham, MA, USA), and Tau (13-6400; Invitrogen, Waltham, MA, USA). Then, the cells were incubated with Alexa 488/594-conjugated anti-mouse or anti-rabbit secondary antibodies (Invitrogen, Waltham, MA, USA). Counterstaining was performed with DAPI (Thermo Fisher Scientific, Waltham, MA, USA). The stained slides were observed via confocal laser-scanning microscopy (FV3000-OSP; Olympus, Tokyo, Japan). 

### 2.10. Whole Staining of Cerebral Organoids

For complete staining of the organoids, the medium and fixed organoids were removed using 4% paraformaldehyde. The fixed organoids were washed with 1× PBS to release the organoid from the Matrigel dome and permeabilized overnight at 4 °C with blocking buffer (5% BSA with 0.5% Triton X-100 in 1× PBS). The organoids were incubated overnight with primary antibodies against TUJ1 (18207; Abcam), MAP2 (13-1500; Invitrogen), and Tau (13-6400; Invitrogen). Then, the cells were incubated overnight with Alexa488/594-conjugated anti-mouse or anti-rabbit secondary antibodies. Counterstaining was performed for 15 min using DAPI. The stained organoids were observed using confocal laser-scanning microscopy. 

### 2.11. Enzyme-Linked Immunosorbent Assay (ELISA) 

A human Aβ42 and Aβ40 ELISA kit (KHB3441, KHB3481; Invitrogen) was used to analyze the Aβ42/40 ratio in the culture supernatant, according to the manufacturer’s instructions. A human Tau ELISA kit (KHB8051, KHB0041; Invitrogen) was used to analyze Tau in the culture supernatant, according to the manufacturer’s instructions. The diluted supernatant was added to each well of the capture antibody-coated plate and incubated for 2 h at room temperature. The detection antibodies were added to each well, incubated for 1 h, and treated with an anti-rabbit IgG horseradish peroxidase solution for 30 min. A stabilized chromogen was added to each well and incubated in the dark for 30 min. The stop solution was added, and the absorbance was detected at 450 nm. 

### 2.12. Statistical Analysis 

The results are expressed as mean and standard error of the mean. Error bars indicate the standard error of the mean. Groups were compared by using Student’s *t*-test and calculating the two-tailed *p* value (* *p* < 0.033, ** *p* < 0.002, *** *p* < 0.001 indicated statistical significance). All statistical analyses were performed by using GraphPad Prism Software 10 (Version 10.2.3, GraphPad, Boston, MA, USA). 

## 3. Results

### 3.1. Reprogramming of PBMCs and Characterization of iPSCs Derived from a Patient with AD

For modeling AD, we generated an iPSC line from the PBMCs of a patient with AD (65, XY) with the E682K mutation in the *APP* gene (21q21.3). The AD iPSCs were generated from a patient with AD using a reprogramming vector containing the Yamanaka factors. The AD iPSCs exhibited a morphology resembling that of human embryonic stem cells and expressed alkaline phosphatase (Figure 1A). Alkaline phosphatase staining revealed that the AD iPSCs were maintained in an undifferentiated state. The AD iPSCs also expressed the pluripotency markers OCT4, SSEA4, TRA-1-60, and TRA-1-81 (Figure 1B). The gene expression of the pluripotency markers (*OCT4*, *NANOG*, *SOX2*, *GABRB3*, and *GDF3*) was confirmed in the AD iPSCs, and the expression levels were compared with those in the H9 human embryonic stem cell line (Figure 1C). The mycoplasma test was negative (Figure 1D). Long-term iPSC culture involves the accumulation of mutations and genomic integration [13]. Karyotyping was performed using the GTG-band method, and the AD iPSCs were confirmed to have a normal karyotype (Figure 1E). To confirm the iPSC differentiation potency, we tested the EB formation and determined the expression of ectoderm (*PAX6*), mesoderm (*ITGA8*), and endoderm (*AFP*) genes in the AD iPSC-derived EBs (Figure 1F,G). To confirm the *APP* gene mutation, we analyzed the DNA sequencing of the HC and AD-iPSCs. Sequencing of the *APP* gene revealed a G to A substitution, which is predicted to result in an amino acid substitution at codon 682 (E682K) (Figure 1H). AD iPSCs were successfully generated from the PBMCs of a patient with AD, and pluripotency was confirmed. Also, the AD-iPSCs reflected the patient’s genetic background of the *APP* gene mutation.

### 3.2. Differentiation of Cerebral Organoids Using iPSCs Derived from a Patient with AD

AD is characterized by a severe loss of neurons in the cortex and certain subcortical regions, followed by cortical dysfunction [14]. For disease modeling, we differentiated COs. To differentiate the iPSC-derived COs, we induced the iPSCs to form EBs in AggreWell plates. A schematic representation of the CO differentiation protocol is shown in Figure 2A. The EBs were transferred to ultra-low attachment plates on day 5 and embedded in Matrigel on day 7. The embedded EBs were maintained in an orbital shaker. The embedded EBs expanded into neuroepithelial cells with the budding of the organoid surface on day 10. On day 20, neural rosettes formed in the organoids. On days 30–40, the morphology of the organoids exhibited alterations, presenting a dense and dark core with cortical layering (Figure 2B). The size of the organoids did not differ from that of the healthy controls (HCs) and patient with AD (Figure 2C). For the histological analysis of the COs, we performed cryosection and immunofluorescence staining. On day 40, the expression of the CO markers TUJ1 and MAP2 increased in the HC and AD iPSC-derived COs (Figure 2D). For a more comprehensive analysis, we performed whole organoid staining. The TUJ1 and MAP2 expressions generally increased in the CO group (Figure 2E). These results confirm that the HC and AD iPSC-derived COs recapitulated the cerebral cortex. 

### 3.3. In Vitro Disease Modeling of AD iPSC-Derived Cerebral Organoids

Increased tau expression is a critical pathological feature in patients with AD; therefore, we analyzed the tau protein levels using cryosection slides (Figure 3A). Tau expression began to increase on day 21 and tended to increase three-fold on day 40 in the AD iPSC-derived COs compared to HC iPSC-derived COs (Figure 3B). Furthermore, we analyzed the whole organoids and confirmed that the tau expression increased in the AD iPSC-derived COs on day 40 (Figure 3C). A fluorescence intensity analysis showed that the expression of tau increased by >6-fold in the AD iPSC-derived COs (Figure 3D). The Aβ42/40 ratio was increased the in AD iPSC-derived COs (Figure 3E). An analysis of tau phosphorylation is essential for understanding its pathology and physiology. The AD iPSC-derived COs showed greater phosphorylated tau levels than the HC iPSC-derived COs in the culture supernatant on day 40 (Figure 3F). These organoids exhibited an increased expression of tau and secretion of phosphorylated tau, confirming the mimicking of the AD pathological phenotype in the AD iPSC-derived COs.

## 4. Discussion

AD is an irreversible and progressive neurodegenerative disease characterized by severe neuronal loss in the cerebral cortex and the accumulation of Aβ and phosphorylated tau proteins [1]. Aβ pathology might involve upstream pathophysiological signals and transfer as an activator of downstream signaling pathways, leading to tau phosphorylation, misfolding, and accumulation [15,16,17]. However, the pathology of tau hyperphosphorylation is unclear, and the exact pathological relationship between tau and Aβ remains unknown. Aβ is a fragment of the transmembrane APP, while tau is a brain-specific microtubule-associated protein. 

The tauopathy of phosphorylation and aggregation is the primary cause of neurodegeneration in AD, leading to neuronal dysfunction and death [18]. The tau pathology can also be found in the brains of patients with very mild dementia without Aβ pathology [19]. Moreover, the approaches to removing Aβ or reducing Aβ production are inadequate for clinical therapy. Thus, interest in tau-based therapies is emerging [20,21]. Experimental models characterized by human tau pathology are necessary to develop tau-based therapies. However, traditional animal models (e.g., transgenic mice) often fail to recapitulate clinical efficacy due to inherent differences in the nervous system complexity, physiology, and drug metabolism between humans and animals [22,23]. The breakthrough was the proposal of patient-derived iPSCs as a promising strategy for disease modeling [7,24,25,26,27,28]. Applying iPSC-derived disease models in drug development can account for differences between animal models and humans. 

HiPSCs can differentiate into the three germ layers and reveal a patient’s genetic background. High-quality undifferentiated patient-derived hiPSCs are essential for disease modeling. Furthermore, it is crucial to verify the identity of cell lines, confirming their undifferentiated state, pluripotency, absence of mycoplasma contamination, karyotype integrity, and capability for differentiation into the three germ layers [29,30]. Poor-quality iPSCs lead to a low differentiation efficiency and faulty analysis. In the Korea National Stem Cell Bank, we confirmed the iPSC quality using their identity, sterility, consistency, stability, and safety to maintain high-quality cell lines. 

Various iPSC-derived disease models have been developed for pathological research and drug screening. HiPSC-derived neuronal disease modeling has traditionally been conducted in a 2D monolayer culture as a simple and low-cost method. However, 2D models are insufficient for mimicking the complex physiology of the human brain. The limitations of 2D models have been overcome by the development of 3D models (e.g., spheroids and organoids). These 3D models more closely recapitulate in vivo environments, cell-to-cell interactions, and neural networks [31] and show increased expressions of neuronal and maturation markers compared to their 2D monolayer counterparts. Furthermore, 3D models effectively replicate tau pathology. Thus, 3D models are a more valuable tool for mimicking neurodegenerative diseases. 

Organoids are self-assembled 3D structures that mimic the functions of human organs. They can be used as mini-organs in disease modeling, drug screening, and personalized medicine. COs recapitulate human brain development and simulate brain function. COs have overcome the differences between human and animal models and emerged as new strategies for neurodegenerative diseases and neurodevelopmental disorders. The functionality of organoids was evaluated through immunofluorescence and immunohistochemical analyses. Traditional cryosections and paraffin sections provide only a partial analysis and incompletely reveal the 3D structure; therefore, we performed a whole analysis of the COs. Whole staining analysis is useful for high-definition structural morphology and localization of overall expression phenotypes. Moreover, this approach eliminates the need for serial sectioning steps, thereby reducing structural damage and saving time compared to cryosectioning or paraffin-sectioning methods. 

Organoids require long-term culture for disease modeling; however, long-term culturing for at least 80 days to 1 year is time-consuming and expensive. In addition, long-term cultures are limited by the lack of nutrients and oxygen in the organoid core. This environment can lead to necrosis within the organoid due to hypoxic conditions. Therefore, improved methods are necessary for efficient disease modeling. 

This study describes a rapid differentiation method for COs, facilitating the efficient modeling of AD. We used an AggreWell plate to produce organoids. The organoids exhibited highly uniform sizes, which could be readily controlled by adjusting the cell density. This resulted in homogenous organoid formation and ensured a high reproducibility for disease modeling purposes. Notably, the expression of neuronal markers was evident by day 14 of differentiation, with a significant increase in tau expression observed in the AD iPSC-derived COs, sustained until day 100. Our method can be used for modeling human tau-overexpressing diseases, particularly aiding in tau-targeted drug screening. Our method provides a shortened differentiation protocol for COs and efficient AD modeling, enabling the effective screening of pharmaceutical compounds and drug development endeavors.

## 5. Conclusions

We generated iPSC lines from a patient with AD harboring the E682K mutation in the *APP* gene. These iPSC lines were then differentiated into COs to model AD pathology. Our findings demonstrated that the AD iPSC-derived COs effectively differentiated within a short time and recapitulated key features of the AD phenotype. Consequently, the AD iPSC-derived CO model holds immediate promise for disease modeling and may increase drug screening feasibility.

## Figures and Tables

**Figure 1 biomedicines-12-01193-f001:**
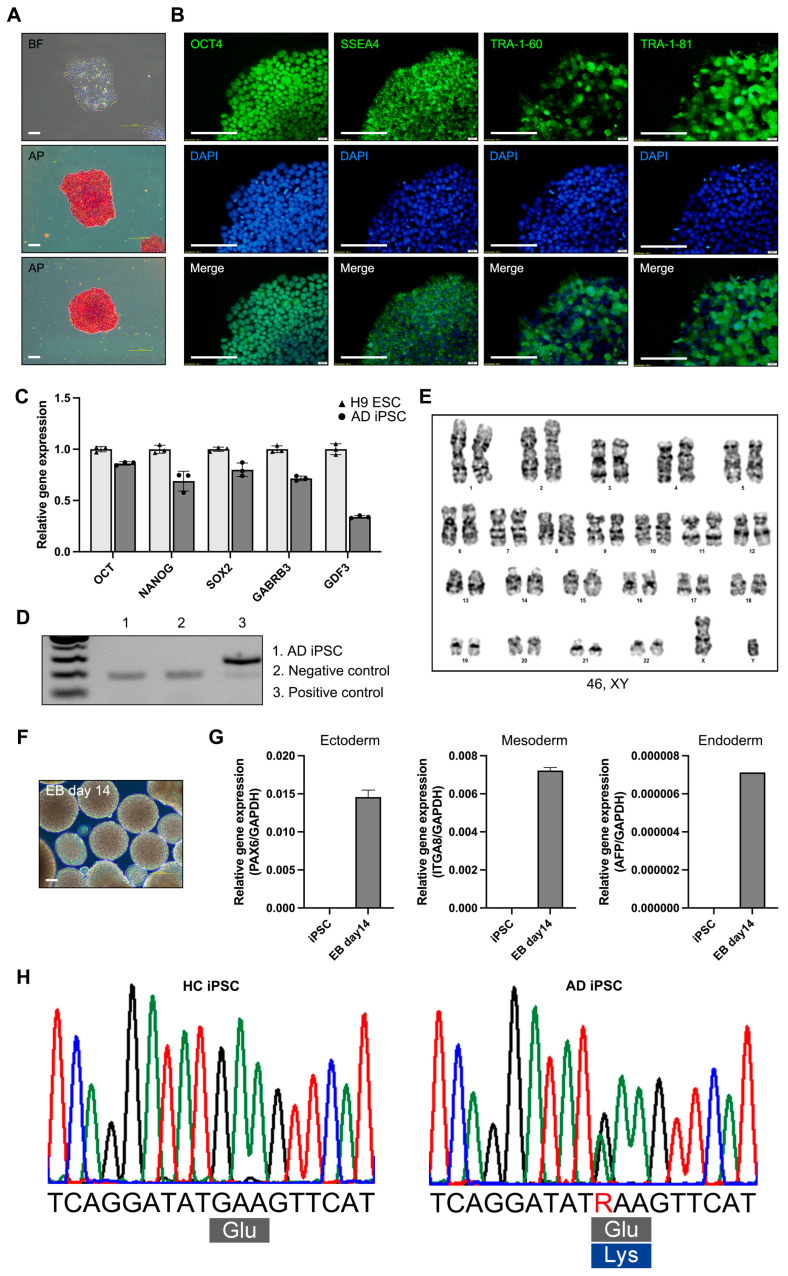
Characterization of induced pluripotent stem cells (iPSCs) derived from a patient with Alzheimer’s disease (AD). (**A**) Morphology and alkaline phosphatase staining of AD iPSC (scale bars: 100 µm). (**B**) Immunofluorescence analysis and DAPI staining of pluripotency markers OCT4, SSEA4, TRA-1-60, and TRA-1-81 (scale bars: 100 µm. (**C**) Real-time polymerase chain reaction (RT-PCR) analysis of the pluripotency marker genes *OCT4*, *NANOG*, *SOX2*, *GABRB3*, and *GDF3*. (**D**) Mycoplasma test of AD iPSC. (**E**) Karyotyping of AD iPSC. (**F**) Morphology of AD iPSC-derived EBs (scale bars: 100 µm). (**G**) RT-PCR analysis of three germ layer differentiation markers: *PAX6* (ectoderm), *ITGA8* (mesoderm), and *AFP* (endoderm). (**H**) Sanger sequencing of HC iPSCs and AD iPSCs. All graphs show the mean and standard error of the mean.

**Figure 2 biomedicines-12-01193-f002:**
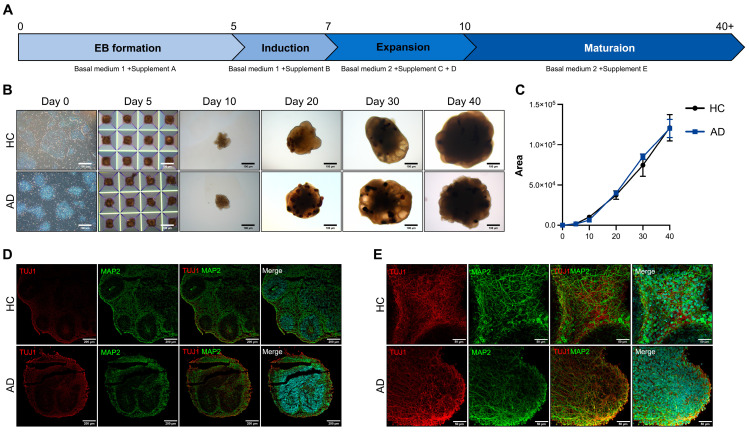
Differentiation of iPSCs into cerebral organoids (COs) and their characterization (**A**) Scheme of the CO differentiation process. (**B**) Morphology of the iPSC-derived COs (scale bars: 100 µm). (**C**) Quantification of the organoid size using the ImageJ program. All graphs show the mean and standard error of the mean. (**D**) Immunofluorescence analysis of TUJ1 and MAP2 with DAPI staining on a cryosection slide (scale bars: 200 µm). (**E**) Whole staining analysis of TUJ1 and MAP2 in COs (scale bars: 50 µm).

**Figure 3 biomedicines-12-01193-f003:**
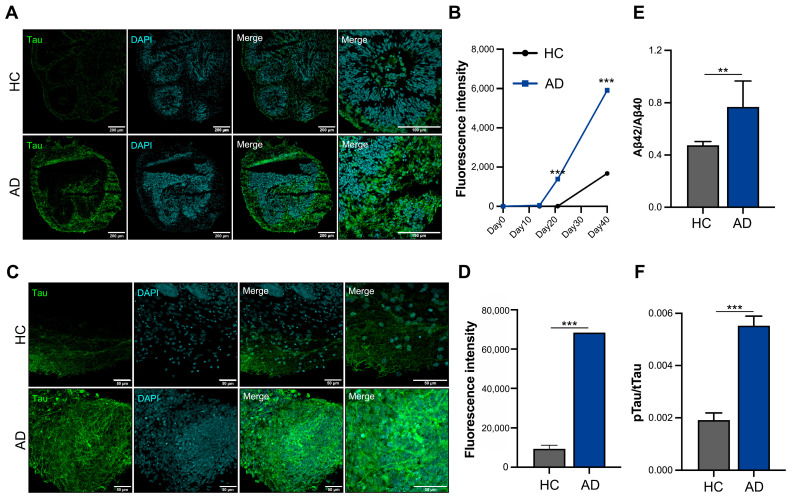
In vitro modeling of Alzheimer’s disease (AD) using iPSCs derived from patient with AD. (**A**) Immunofluorescence staining of cryosectioned tissue. (**B**) Quantification of the fluorescence intensity of tau using cellSens imaging software (version 4.1, Olympus). (**C**) Whole staining analysis of cerebral organoids (COs) (scale bars: 50 µm). (**D**) Quantification of fluorescence intensity of tau in whole organoids. (**E**) Aβ42/40 ratio analysis in the culture supernatant. (**F**) Phosphorylation analysis of the tau protein in the culture supernatant. All graphs show the mean and standard error of the mean (** *p* < 0.002, *** *p* < 0.001, as determined by Student’s *t*-test).

**Table 1 biomedicines-12-01193-t001:** Clinical information of a healthy control and patient with Alzheimer’s disease.

No	Disease	Sex	Age	Gene	Mutation	Symptom
1	Healthy control	M	50	-	-	-
2	Alzheimer’s disease	M	65	*APP*	c.2044G>A (p.E682K)	Amyloid positive

**Table 2 biomedicines-12-01193-t002:** TaqMan® probe ID and primer list.

Analysis	Target	Length	Primer Sequence
Pluripotency markers	*OCT4*	65 bp	TaqMan Probe ID Hs00742896-s1
	*NANOG*	109 bp	TaqMan Probe ID Hs02387400-g1
	*SOX2*	121 bp	TaqMan Probe ID Hs00602736-s1
	*GABRB3*	63 bp	TaqMan Probe ID Hs00241459-m1
	*GDF3*	65 bp	TaqMan Probe ID Hs00220998-m1
Housekeeping gene	*GAPDH*	122 bp	TaqMan Probe ID Hs999999905-m1
Three germ layer differentiation markers	*PAX6*	76 bp	TaqMan Probe ID Hs00240871-m1
	*ITGA8*	89 bp	TaqMan Probe ID Hs00233321-m1
	*AFP*	82 bp	TaqMan Probe ID Hs00173490-m1
Housekeeping gene	*GAPDH*	58 bp	TaqMan Probe ID Hs03929097-g1
Targeted mutation analysis/Sanger sequencing	*APP-F1*	24-mer	CAGGCCTAGAAAGAAGTTTTGGGT
	*APP-F2*	21-mer	CTAATTGGTTGTCCTGCATAC
	*APP-R1*	21-mer	GATTTCTAGCACAGGATGAAC
	*APP-R2*	22-mer	GATGAACCAGAGTTAATAGGTC

## Data Availability

All datasets of this article are included within the article.

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
