# Peer review of "Generation of Alzheimer’s Disease Model Derived from Induced Pluripotent Stem Cells with APP Gene Mutation"

_biomedicines, 2024, doi:10.3390/biomedicines12061193_

Round 1

Reviewer 1 Report

Comments and Suggestions for Authors

Kim et al present a study on the generation of a novel iPSC line from an individual with an E682K mutation in APP. They characterise the stem cells and then generate cerebral organoids, demonstrating an increase in tau levels.

This is a robust study and generates a tool that is useful for the community and of general interest to the readership of biomedicines. I would be interested to hear the authors’ thoughts on the following points. 

1. Is it possible to show Sanger sequencing data to confirm the mutation in the iPSCs. 

2. I am unclear how many patient-derived iPSC lines have been generated - “patients” line 69. Please can you give detailed information of the donor/donors (age at disease onset, age at biopsy, gender, APOE status etc, if available). Are the HC organoids from the H9 line? Please add these details to the methods. 

3. It is important to mention how many batches of COs were made? Do the results represent more than one batch of organoids? Replication is crucial due to the bath-to-batch variability of CO protocols (data in Fig2C and Fig 3B, 3D, 3E). 

4. Did the authors consider measuring Abeta levels? (Note this study PMID 21500352). 

5. The authors describe a shortened protocol (line 70 and 295). Please can you clarify what this is compared to? It seems that the StemDiff kit has been used, which is a direct parallel to the Lancaster protocols. Also, are the numbers in the methods correct – 900 cells (line 123), 1ml media (line 124). 

6. Please state the timepoints used Figure 3, i.e. at what age is are the organoids. 

Reviewer 2 Report

Comments and Suggestions for Authors

In this work, the authors generated cerebral ogranoids (CO) using patient-derived induced pluripotent stem cells (iPSCs) created from peripheral blood mononuclear cells (PBMCs) of individual with Alzheimer’s disease (AD) carrying E682K mutation in the gene encoding amyloid precursor protein (APP). Such organoids can be used as a model of AD.

Please find my comments below.

Line 18. “in the amyloid precursor protein (APP) gene” - it's better to say “in the gene, encoding the amyloid precursor protein (APP)”

Line 36. “caused by gene mutations in amyloid precursor protein (APP) gene” - in my opinion, it is a redundancy/tautology (I mean, to say “gene munations in the gene”), Why do not you say “caused by mutations in APP gene, encoding amyloid precursor protein (APP)”?

Question to the authors – did you use PBMCs from multiple donors (as stated in the text of the manuscript) or from only one 65 years patient? See line167 “an iPSC line from PBMCs of patients with AD (65, XY) with the E682K mutation in the APP gene (21q21.3)”.

Why do you refer to one patient as “patients”?

Also, gene names should be italicized (for example, APP gene), while protein names should not.

As for Real-time polymerase chain reaction (RT-PCR) analysis of AD iPSC, I suggest more accurately choosing housekeeping genes for it. There are “iPSCs-specific” housekeeping genes that do not change during all stages of differentiation/de-differentiaion. Also, its advisable to use several housekeeping genes simultaneously in one RT-PCR reaction, rather than one.

Line 120. “anti-adherence rinsing solution” - please specify the chemical composition of this solution or provide a catalog number and manufacturer info.

The AggreWell plates were incubated at 37°C”. and 5% CO2 perhaps?

As the protocol for reprogramming of PBMCs, maintenance of iPSCs and their subsequent differentiation are the main messages of the article, could you please in the discussion of the manuscript elaborate on the details of the protocol, comparing it to other similar approaches?

Author Response

Detailed Responses to Reviewers

May 18, 2024

  • Journal: Biomedicines
  • Manuscript #: biomedicines-3015583
  • Title of Paper: Generation of an Alzheimer’s disease model derived from induced pluripotent stem cells with an APP gene mutation
  • Authors: Yena Kim (First author); Binna Yun; Byoung Seok Ye; Bo-Young Kim (Corresponding author)

Dear Assistant Editor:  

We wish to re-submit the attached manuscript, entitled “Generation of an Alzheimer’s disease model derived from induced pluripotent stem cells with an APP gene mutation” in Biomedicines. The manuscript ID is biomedicines-3015583. Thank you for the opportunity to revise and resubmit this manuscript. We have carefully reviewed your comments and have revised our manuscript based on your suggestions. We appreciate the time, efforts, and the constructive feedback provided by the editor and reviewers in reviewing this manuscript.

We have addressed each suggestion specifically, and our responses are listed in a point-by-point manner below. Changes to the manuscript are highlighted in yellow.

Thank you very much for your consideration of our manuscript. We look forward to hearing from you.

Sincerely,

Bo-Young Kim, Ph.D

Division of Intractable Diseases Research, Department of Chronic Diseases Convergence Research, Korea National Institute of Health

Reviewer: 2

In this work, the authors generated cerebral organoids (CO) using patient-derived induced pluripotent stem cells (iPSCs) created from peripheral blood mononuclear cells (PBMCs) of individual with Alzheimer’s disease (AD) carrying E682K mutation in the gene encoding amyloid precursor protein (APP). Such organoids can be used as a model of AD.

  1. Line 18. “in the amyloid precursor protein (APP) gene” – it’s better to say “in the gene, encoding the amyloid precursor protein (APP)”. Line 36. “caused by gene mutations in amyloid precursor protein (APP) gene”- in my opinion, it is a redundancy/tautology (I mean, to say “gene mutations in the gene”), Why do not you say “caused by mutations in APP gene, encoding amyloid precursor protein (APP)”?

Response: Thank you for your evaluating our manuscript and for the insightful comments. Based on your comments, we have revised these sentences to avoid confusion.

Line 20: “…in the gene, encoding the amyloid precursor protein.”

Lines 36–37: “…is caused by mutations in APP gene, encoding amyloid precursor protein (APP).” 

  1. Did you use PBMCs from multiple donors(as stated in the text of the manuscript) or from only one 65 years patient? See line 167 “an iPSC line from PBMCs of patients with AD (65,XY) with the E682K mutation in the APP gene (21q21.3)”. Why do you refer to one patient as “patients”? Also, gene names should be italicized (for example, APP gene), while protein names should not.

Response: Thank you for your kind comment. As you mentioned, we used one patient and have made the necessary edits in our revised manuscript.

  1. As for Real-time polymerase chain reaction(RT-PCR) analysis of AD iPSC, I suggest more accurately choosing housekeeping genes for it. There are “iPSCs-specific” housekeeping genes that do not change during all stages of differentiation/de-differentiation. Also, its advisable to use several housekeeping genes simultaneously in one RT-PCR reaction, rather than one.

Response: Thank you for your generous comment. We used a commercially available primer from Thermo Fisher Scientific for RT-PCR. This primer is widely used, not only by our group but also by many other groups [1-4]. TaqMan probe primers are designed using a highly sophisticated oligonucleotide probe/primer design pipeline, which includes robust primer design algorithms and an extensive array of bioinformatics tools and processes to automate assay design. We selected one of these primers and conducted the RT-PCR examination accordingly.

  1. Line 120. “anti-adherence rinsing solution”-please specify the chemical composition of this solution or provide a catalog number and manufacturer info. “The Aggrrewell plates were incubated at 37°C and 5% CO2 perhaps?.

Response: Thank you for your comment. We used the anti-adherence rinsing solution of stem cell technologies, and AggreWell plates were incubated at 37°C with 5% CO2. We reflected your comments and edited the Materials and Methods section.

  1. As the protocol for reprogramming of PBMCs, maintenance of iPSCs and their subsequent differentiation are the main messages of the article, could you please in the discussion of the manuscript elaborate on the details of the protocol, comparing it to other similar approaches?

Response: Thank you for your sharply comment. Based on your comments, we have added the following part to the Discussion section:

“We used AggreWell plate to generate organoids. The size of the organoids is highly uniform and can be easily controlled by adjusting the cell density, leading to homogenous organoid and high reproducibility for disease modeling.” (Lines 306–308)

References

  1. Xu, Y.; Takahashi, Y.; Wang, Y.; Hama, A.; Nishio, N.; Muramatsu, H.; Tanaka, M.; Yoshida, N.; Villalobos, I.B.; Yagasaki, H.; et al. Downregulation of GATA-2 and overexpression of adipogenic gene-PPARgamma in mesenchymal stem cells from patients with aplastic anemia. Exp Hematol 2009, 37, 1393-1399, doi:10.1016/j.exphem.2009.09.005.
  2. Ma, Y.H.; Mentlein, R.; Knerlich, F.; Kruse, M.L.; Mehdorn, H.M.; Held-Feindt, J. Expression of stem cell markers in human astrocytomas of different WHO grades. J Neurooncol 2008, 86, 31-45, doi:10.1007/s11060-007-9439-7.
  3. Stich, S.; Haag, M.; Haupl, T.; Sezer, O.; Notter, M.; Kaps, C.; Sittinger, M.; Ringe, J. Gene expression profiling of human mesenchymal stem cells chemotactically induced with CXCL12. Cell Tissue Res 2009, 336, 225-236, doi:10.1007/s00441-009-0768-z.
  4. Park, T.S.; Galic, Z.; Conway, A.E.; Lindgren, A.; van Handel, B.J.; Magnusson, M.; Richter, L.; Teitell, M.A.; Mikkola, H.K.; Lowry, W.E.; et al. Derivation of primordial germ cells from human embryonic and induced pluripotent stem cells is significantly improved by coculture with human fetal gonadal cells. Stem Cells 2009, 27, 783-795, doi:10.1002/stem.13.

---------------------------------------------------------------------------------------------------------------

The English in this document has been checked by at least two professional editors, both native speakers of English. For a certificate, please see:

http://www.editage.co.kr